# OpenReview forum: "CIMemories: A Compositional Benchmark For Contextual Integrity In LLMs"
_ICLR.cc/2026/Conference — ICLR 2026 Poster_

### Official Review · Reviewer_9MtN · 2025-10-27

**Soundness:** 3
**Presentation:** 3
**Contribution:** 3
**Rating:** 6
**Confidence:** 4

**Summary:**

This work introduces CIMemories, a benchmark for testing whether memory-augmented LLMs respect Contextual Integrity such that LLMs reveal stored user facts to third parties only when appropriate, while still being helpful. It composes synthetic user profiles and social contexts so the same attribute can be required in one setting but inappropriate in another, and evaluates models with two complementary measures: violation (leaking “not-to-share” facts) and completeness (sharing what’s needed). Experiments on frontier models reveal a clear privacy–utility trade-off and accumulating leaks across tasks, with scaling and “reasoning” prompts offering only modest relief. The contribution is a clear formalisation of Contextual Integrity, a controllable compositional dataset, and an empirical study that identifies where current assistants fail and how mitigation shifts the trade-offs.

**Strengths:**

* The paper applies Contextual Integrity (CI) to memory-augmented LLMs in a compositional setting, formalising a benchmark where the same attributes of a user profile can be appropriate in one context but inappropriate in another. The ideas of flexible memory composition and multi-task composition per user are clearly specified and expand the scope beyond single-instance evaluations. The two scores, Violation@n and Completeness, provide a simple but representative evaluation of the privacy–utility trade-off in persistent LLM memory

* The methodology is rigorous and systematic. The paper describes the data-generation pipeline, the evaluation setup, and uses quantitative and qualitative analyses across multiple models and configurations. Results include consistent patterns (e.g., violation/completeness trade-off, scaling saturation, and domain-level “granularity” errors), supported by tables/figures and concrete violation excerpts.

* The presentation is clear. The benchmark workflow, task formation, and the REVEAL judge are explicitly laid out, with prompt templates and an overview figure that maps profiles, contexts, metrics, and the judge’s role. This makes it straightforward to understand what counts as an explicit reveal and how metrics are computed

* The work addresses a timely risk in persistent memory: disclosing the information in the wrong place. By quantifying violations and completeness, showing accumulation over tasks/generations, and analyzing scaling, reasoning, and conservative prompts, it offers actionable insights for deployment and motivates future mitigation work.

**Weaknesses:**

* Reliance on LLM-Generated Ground Truth and Judges: The benchmark’s contextual integrity labels are entirely generated using closed-source LLMs (GPT-5 personas) and judged by another LLM (DeepSeek-R1) that only flags explicit disclosures. This creates a potential circular evaluation loop, the same class of models(GPT-5) being tested also defines the “ground truth.” To strengthen validity, the authors can include a small-scale human validation study to measure inter-annotator agreement and human–LLM alignment.

* Limited Coverage and Cultural Bias in Labeling: The dataset includes only 10 user profiles and further filters out all attribute–context pairs where privacy personas disagree (entropy > 0), resulting in potential selection bias toward “easy-to-label” cases. Moreover, the benchmark grounds its contextual-integrity labels in Westin’s U.S. privacy personas and a set of U.S.-centric contexts (e.g., HR departments, IRS agents, USCIS officers). Because contextual-integrity norms vary across cultures, this narrow framing limits the benchmark’s representativeness and generalizability. Expanding the dataset with cross-cultural personas, diverse contexts, and non-U.S. human raters would both mitigate coverage bias and improve external validity for global adoption.

* Minor Presentation Issues: A dangling “?” in § 5.2.2.

**Questions:**

* What proportion of attribute–context pairs survive the unanimity filter across Westin personas (entropy = 0), and how are discarded pairs distributed across domains?

* Could you run a small human validation study to measure inter-annotator agreement and human–LLM alignment on “share” vs “private” labels? This would help verify that LLM-generated CI labels reflect plausible human norms.

* Since CI norms vary culturally, and current personas and contexts are U.S.-centric (e.g., IRS, USCIS), would you consider adding cross-cultural personas and context to evaluate whether violation/completeness patterns shift in future work?

---

> ### Author Response · Authors · 2025-11-21
> **Response by Authors**
>
> We sincerely thank the reviewer for their comments and suggestions, and provide a detailed discussion below:
>
> >Reliance on LLM-Generated Ground Truth and Judges: The benchmark’s contextual integrity labels are entirely generated using closed-source LLMs (GPT-5 personas) and judged by another LLM (DeepSeek-R1) that only flags explicit disclosures. This creates a potential circular evaluation loop, the same class of models(GPT-5) being tested also defines the “ground truth.” To strengthen validity, the authors can include a small-scale human validation study to measure inter-annotator agreement and human–LLM alignment.
>
> >Could you run a small human validation study to measure inter-annotator agreement and human–LLM alignment on “share” vs “private” labels? This would help verify that LLM-generated CI labels reflect plausible human norms.
>
> As suggested, we perform a brief human validation study to compute agreement. We randomly sample a total of 100 attribute-context pairs (50 not-to-share, and 50 to-share as labelled by GPT-5 personas), and ask 3 human raters (2 authors, 1 non-author) to label them. We then compute agreement between each rater and the model's labels, as well as between raters as below. Overall, we find that model predicts human labels respectably:
>
> |             | Model | Rater 1 | Rater 2 | Rater 3 |
> | ----------- | ----- | ------- | ------- | ------- |
> | **Model**   | -     | 85%     | 87%     | 83%     |
> | **Rater 1** | -     | -       | 83%     | 84%     |
> | **Rater 2** | -     | -       | -       | 80%     |
> | **Rater 3** | -     | -       | -       | -       |

---

> > ### Author Response · Authors · 2025-11-21
> > **Response by Authors (cont.)**
> >
> > >Limited Coverage and Cultural Bias in Labeling: The dataset includes only 10 user profiles and further filters out all attribute–context pairs where privacy personas disagree (entropy > 0), resulting in potential selection bias toward “easy-to-label” cases. Moreover, the benchmark grounds its contextual-integrity labels in Westin’s U.S. privacy personas and a set of U.S.-centric contexts (e.g., HR departments, IRS agents, USCIS officers). Because contextual-integrity norms vary across cultures, this narrow framing limits the benchmark’s representativeness and generalizability. Expanding the dataset with cross-cultural personas, diverse contexts, and non-U.S. human raters would both mitigate coverage bias and improve external validity for global adoption.
> >
> > >Since CI norms vary culturally, and current personas and contexts are U.S.-centric (e.g., IRS, USCIS), would you consider adding cross-cultural personas and context to evaluate whether violation/completeness patterns shift in future work?
> >
> > We agree with the reviewer that increasing representativeness of the benchmark is a excellent direction to capturing a wide variety of norms. We will add a discussion to the paper of how, as part of future work, an open problem is to augment the benchmark by including full human labelling via crowdsourcing human raters, selected to capture a wide geographic, demographic, and cross-cultural representation. We also plan to pursue this ourselves in future work. Another discussion point we will add is on developing more cross-cultural contexts, achievable through more intricate synthetic data generation seeds. In general, we also plan to keeping the benchmark continuously updated.
> >
> > We have also added experiments with an increased number of user profiles (an additional 60), and for each of the persona-wise labels to provide some insight on the cases where privacy personas disagree:
> >
> > More profiles:
> >
> > | Model                  | Violations@n | Completeness |
> > | ---------------------- | ------------ | ------------ |
> > | GPT-5                  | 26.2%        | 53.8%        |
> > | o3                     | 34.8%        | 50.5%        |
> > | GPT-4o                 | 16.7%        | 40.1%        |
> > | Gemini 2.5 Flash       | 49.0%        | 49.4%        |
> > | Llama-3.3 70B Instruct | 42.8%        | 48.7%        |
> > | Qwen-3 32B             | 65.1%        | 53.4%        |
> >
> > Privacy Fundamentalist:
> >
> > | Model                    | Violations@n | Completeness |
> > | ------------------------ | ------------ | ------------ |
> > | GPT-5                    | 42.1%        | 52.8%        |
> > | o3                       | 57.7%        | 51.9%        |
> > | GPT-4o                   | 30.9%        | 40.7%        |
> > | Gemini 2.5 Flash         | 64.3%        | 50.4%        |
> > | Llama-3.3 70B Instruct   | 59.7%        | 51.2%        |
> > | Qwen-3 32B               | 81.4%        | 55.4%        |
> > | Claude-4 Sonnet          | 59.1%        | 56.9%        |
> > | Mistral-7B Instruct v0.3 | 73.0%        | 43.4%        |
> >
> > The Pragmatic:
> >
> > | Model                    | Violations@n | Completeness |
> > | ------------------------ | ------------ | ------------ |
> > | GPT-5                    | 34.3%        | 48.3%        |
> > | o3                       | 50.6%        | 47.6%        |
> > | GPT-4o                   | 23.1%        | 35.9%        |
> > | Gemini 2.5 Flash         | 57.0%        | 45.7%        |
> > | Llama-3.3 70B Instruct   | 56.3%        | 46.2%        |
> > | Qwen-3 32B               | 77.6%        | 50.9%        |
> > | Claude-4 Sonnet          | 54.6%        | 52.6%        |
> > | Mistral-7B Instruct v0.3 | 65.8%        | 39.5%        |
> >
> >
> > The Unconcerned:
> >
> > | Model                   | Violations@n | Completeness |
> > | ----------------------- | ------------ | ------------ |
> > | GPT-5                   | 34.9%        | 44.6%        |
> > | o3                      | 45.5%        | 43.1%        |
> > | GPT-4o                  | 20.3%        | 31.4%        |
> > | Gemini 2.5 Flash        | 56.9%        | 41.5%        |
> > | Llama-3.3 70B Instruct  | 52.4%        | 42.0%        |
> > | Qwen-3 32B              | 75.2%        | 47.2%        |
> > | Claude-4 Sonnet         | 51.6%        | 48.7%        |
> > | Mistal-7B Instruct v0.3 | 62.5%        | 36.3%        |
> >
> >
> >
> >
> > >Minor Presentation Issues: A dangling “?” in § 5.2.2.
> >
> > We appreciate the pointer, and will fix this.
> >
> > >What proportion of attribute–context pairs survive the unanimity filter across Westin personas (entropy = 0), and how are discarded pairs distributed across domains?
> >
> > Roughly 31% of attribute-context pairs were discarded during the unanimity filter, distributed across the domains as: employment (16.23%), housing (10.29%), legal (10.19%), health (9.83%), schedule (9.49%), general (9.41%), mental_health (9.25%), finance (8.83%), education (8.29%), relationships (8.19%).

---

### Official Review · Reviewer_7wCV · 2025-10-28

**Soundness:** 3
**Presentation:** 3
**Contribution:** 3
**Rating:** 8
**Confidence:** 4

**Summary:**

The paper proposed a new benchmark focusing on evaluating how LLMs correctly leverages in-context memories, particularly user profiles. The problem is that, based on different tasks, the LLM should reveal certain user information but not others. The paper curated a new benchmark which features two key innovations: (1) flexible memory composition; (2) multi-task composition; The results show that recent LLMs still struggle with many violations on user privacy and shows a trade-off between task completeness and contextual integrety

**Strengths:**

- The problem definition and motivation is very clear and under-explored in the community
- The paper writing and techical details are clear; The analysis on the violation vs completeness trade-off, and the impact of model size and thinking is very helpful for further understanding the challenge

**Weaknesses:**

- The discussion on how to mitigate the problem is too weak. It is ideal to have an initial improved baseline based on the insights from the benchmarking analysis; these actionable insights are most helpful to the community; for example, which type of reasoning may benefit most to reduce violation while keeping completeness and general performance?
- Without a quantative comparison with prior contextual privacy benchmarks, it is unclear whether the CIMemories benchmark is testing similar skills or is actually revealing some new challenges. Concretely, it would be good to add columns in Table 1 reflecting performance in prior privacy related benchmarks such as CI-Bench, to show if there is a strong correlation between the performances.
- Minor: The use of color in Figure 4, 5 can be improved considering red-green color blind readers

**Questions:**

- Figure 5 (b) seems to show that reasoning is a promising direction in addressing this tradeoff; can the authors provide more details on how the reasoning is performed in the experiments

---

> ### Author Response · Authors · 2025-11-21
> **Response by Authors**
>
> We sincerely thank the reviewer for their comments and suggestions, and provide a detailed discussion below:
>
>
>
> >The discussion on how to mitigate the problem is too weak. It is ideal to have an initial improved baseline based on the insights from the benchmarking analysis; these actionable insights are most helpful to the community; for example, which type of reasoning may benefit most to reduce violation while keeping completeness and general performance?
>
> To better understand the explicit type of reasoning that can reduce violations, we sample 100 responses from the Qwen3-30B Thinking variant, and manually inspect the reasoning traces. Here, we find an extremely common theme of the model *explicitly enumerating full attribute/groups of attributes to not include, along with justification*. In some cases, this is as explicit as a bulleted list. For example, in a message to be sent to a defense attorney, we observe the following reasoning:
>
> *Red flags to avoid*:
> - Don't say "I'm stressed" - focus on facts
> - Don't mention spouse's support (irrelevant to strategy)
> - Skip home maintenance costs ($1,200) - not legally material
>
> In most cases, we see some form of blanket "irrelevance" as the model's justification for not including some information. This likely captures many egregious, obvious violations (and hence the improvements over non-thinking). We conjecture that perhaps one way to improve this even further would be reasoning that can enumerate the downstream consequences of sharing each attribute. We will certainly add a discussion of this to the paper.
>
>
> >Without a quantative comparison with prior contextual privacy benchmarks, it is unclear whether the CIMemories benchmark is testing similar skills or is actually revealing some new challenges. Concretely, it would be good to add columns in Table 1 reflecting performance in prior privacy related benchmarks such as CI-Bench, to show if there is a strong correlation between the performances.
>
> While CI-Bench [1] has not been publicly released, we evaluate on the Confaide benchmark, which has been released [2], where the model must construct actions-items and summaries from meetings transcripts where a secret has been discussed. We run Confaide using GPT-5, and find that GPT-5 achieves exactly 0% violation (with completeness of 42%), demonstrating that CIMemories uncovers new contextual integrity failure modes of frontier models. We will include these results in the paper.
>
> In general, a key difference of CIMemories from previous benchmarks is to test model capabilities when handling multiple information attributes, where some attributes should be shared in particular contexts but not in others. In contrast, prior benchmarks are largely focused on a single information attribute to share/not-to-share.
>
> [1] Cheng, Zhao, et al. "Ci-bench: Benchmarking contextual integrity of ai assistants on synthetic data." _arXiv preprint arXiv:2409.13903_ (2024).
>
> [2] Mireshghallah, Niloofar, et al. "Can LLMs Keep a Secret? Testing Privacy Implications of Language Models via Contextual Integrity Theory." _The Twelfth International Conference on Learning Representations_.
>
> >Minor: The use of color in Figure 4, 5 can be improved considering red-green color blind readers
>
> We appreciate this pointer, and will certainly redo this figure with non-red/green colors.
>
> >Figure 5 (b) seems to show that reasoning is a promising direction in addressing this tradeoff; can the authors provide more details on how the reasoning is performed in the experiments
>
> To evaluate reasoning in the experiments, we use the Qwen3-30B-Thinking variant, which always begins its response with a “reasoning chain”, which is concretely a long text sequence enclosed in \<think> and \</think> tokens. No additional (i.e., no "step-by-step") prompting is supplied to the model. Per the model authors, this model is obtained by starting with a “hybrid” model (Qwen3-30B) that allows the user to switch between reasoning and non-reasoning by beginning or not beginning the response with a \<think> token, and is likely trained on traditional tasks such as math, coding, logic, creative writing, etc. They then further train (presumably more CoT SFT and/or some reasoning RL) to yield a separate “thinking-only” model variant that always begins its response with \<think>, and a separate “instruct-only” model that does not include \<think>\</think> at all. The final model response for the “thinking-only” variant is parsed as all text outside the \<think>\</think> block.

---

### Official Review · Reviewer_D8mp · 2025-10-31

**Soundness:** 3
**Presentation:** 2
**Contribution:** 2
**Rating:** 4
**Confidence:** 3

**Summary:**

This paper introduces CIMemories, a benchmark for evaluating whether memory-augmented LLMs appropriately control information flow based on social context, based on Contextual Integrity theory. The benchmark features synthetic user profiles with ~147 attributes each across 9 domains (finance, health, employment, etc.), paired with ~45 task contexts per user. It evaluates 8 models and find violation rates of 14-69%.  It also demonstrates that violations accumulate over time as users engage in more tasks, and that current mitigation strategies (scaling, prompting) provide limited relief.

**Strengths:**

S1. Timely and critical benchmark. Most of the companies are deploying Memory-augmented LLMs, but prior benchmarks don't capture the compositional nature of contextual privacy.

S2: The formalization in Section 3 clearly connects CI theory to measurable metrics, with appropriate violation and completeness perspectives.

S3. Analysis is done properly. The granularity failure finding (models identify right domain but over-share details) and the domain-wise breakdown in Figure 3 provides interpretable insights.

**Weaknesses:**

W1. Benchmark scale is one of the major concern that I have. With only 10 profiles and no statistical testing, the results lack the rigor needed for definitive conclusions.

W2. Synthetic data. While acknowledged, this is a fundamental limitation. Real users have complex, inconsistent preferences that synthetic profiles cannot capture.

W3. Requiring unanimous agreement across all 3 personas discards many valid scenarios. This may bias toward only "obvious" privacy violations. Real privacy often involves legitimate disagreement, which is excluded and the paper doesn't report what percentage of attribute-context pairs were discarded.

**Questions:**

Q1: 10 profiles is very less in my pov to make result significant. I get the cost but can it be increased for the open-source models where inference is cheaper?

Q2. Synthetic data might not capture complex real user preference. Real LLM users have maybe 20-50 memories after a few months. 147 attributes per profile seems to high. Why so many? Any plan to collect real user data?

Q3. What percentage of attribute-context pairs were discarded?

---

> ### Author Response · Authors · 2025-11-21
> **Response by Authors**
>
> We sincerely thank the reviewer for their comments and suggestions, and provide a detailed discussion below:
>
>
> >W1. Benchmark scale is one of the major concern that I have. With only 10 profiles and no statistical testing, the results lack the rigor needed for definitive conclusions.
>
> >Q1: 10 profiles is very less in my pov to make result significant. I get the cost but can it be increased for the open-source models where inference is cheaper?
>
> We had restricted ourselves to 10 profiles primarily so that those running the benchmark can do so on frontier closed-source models without incurring significant cost, and because generating labels also incurs significant cost from GPT-5. In general, the profiles are an abstraction for a collection of attribute-context pairs over which violations and completeness are measured, for which we have over 40K pairs. However, we agree that adding more profiles could provide for an extended version of the benchmark. To this end, we have generated an additional 60 profiles. We have evaluated on them below, and find the results to be similar and model rankings stay the same. We will certainly add these results to the paper.
>
> | Model                  | Violations@n | Completeness |
> | ---------------------- | ------------ | ------------ |
> | GPT-5                  | 26.2%        | 53.8%        |
> | o3                     | 34.8%        | 50.5%        |
> | GPT-4o                 | 16.7%        | 40.1%        |
> | Gemini 2.5 Flash       | 49.0%        | 49.4%        |
> | Llama-3.3 70B Instruct | 42.8%        | 48.7%        |
> | Qwen-3 32B             | 65.1%        | 53.4%        |
>
>
> >W2. Synthetic data. While acknowledged, this is a fundamental limitation. Real users have complex, inconsistent preferences that synthetic profiles cannot capture.
>
> >Q2. Synthetic data might not capture complex real user preference. Real LLM users have maybe 20-50 memories after a few months. 147 attributes per profile seems to high. Why so many? Any plan to collect real user data?
>
> We agree that there will be a gap between synthetic and true user data. However, many of our design decisions behind CIMemories (such as constructing seeds for events and information domains) are drawn from real studies on humans secrecy [1], to capture real users as much as possible. Another key challenge we faced while crafting such a benchmark is obtaining such data about real users, to be released as a dataset to the community without violating their privacy. In particular, in many cases we would like to study sensitive attributes (e.g., those regarding relationships, or medical conditions), but constructing such a dataset and releasing could violate the privacy of the real users.
>
> Regarding the number of attributes, we were trying to study the contextual privacy properties of models in the fully-personalized setting, where a large amount of user information has been incorporated as memories. However, one advantage of our CIMemories is that one can also use it to also study settings with fewer attributes/memories by only placing this smaller subset into the memory of the model (e.g., fewer events, or less attributes per event). For example, we perform such an analysis in Figure 4 of the paper, which studies the evolution of violations and completeness as memories accumulate over time, from very few to dozens.
>
> [1] McDonald, Rachel I., et al. "Motivated secrecy: Politics, relationships, and regrets." _Motivation Science_ 6.1 (2020): 61.

---

> > ### Author Response · Authors · 2025-11-21
> > **Response by Authors (cont.)**
> >
> > >W3. Requiring unanimous agreement across all 3 personas discards many valid scenarios. This may bias toward only "obvious" privacy violations. Real privacy often involves legitimate disagreement, which is excluded and the paper doesn't report what percentage of attribute-context pairs were discarded.
> >
> > >Q3. What percentage of attribute-context pairs were discarded?
> >
> > We agree with the reviewer that the cases where personas disagree are more interesting from a privacy discussion point of view. However, these cases are where the privacy measures become personal, and we wanted to avoid biasing towards any specific group of users. For our main results we mainly focused on estimating (a lower bound on) privacy violations that the population would agree upon. Indeed, on a per-user basis, the violations could be more depending upon their preferences - during the unanimous agreement filter, roughly 31% of attribute-context pairs were discarded. One way to help provide insight into these scenarios is to add results for each of the persona-wise labels. We provide these below:
> >
> > Privacy Fundamentalist:
> >
> > | Model                    | Violations@n | Completeness |
> > | ------------------------ | ------------ | ------------ |
> > | GPT-5                    | 42.1%        | 52.8%        |
> > | o3                       | 57.7%        | 51.9%        |
> > | GPT-4o                   | 30.9%        | 40.7%        |
> > | Gemini 2.5 Flash         | 64.3%        | 50.4%        |
> > | Llama-3.3 70B Instruct   | 59.7%        | 51.2%        |
> > | Qwen-3 32B               | 81.4%        | 55.4%        |
> > | Claude-4 Sonnet          | 59.1%        | 56.9%        |
> > | Mistral-7B Instruct v0.3 | 73.0%        | 43.4%        |
> >
> > The Pragmatic:
> >
> > | Model                    | Violations@n | Completeness |
> > | ------------------------ | ------------ | ------------ |
> > | GPT-5                    | 34.3%        | 48.3%        |
> > | o3                       | 50.6%        | 47.6%        |
> > | GPT-4o                   | 23.1%        | 35.9%        |
> > | Gemini 2.5 Flash         | 57.0%        | 45.7%        |
> > | Llama-3.3 70B Instruct   | 56.3%        | 46.2%        |
> > | Qwen-3 32B               | 77.6%        | 50.9%        |
> > | Claude-4 Sonnet          | 54.6%        | 52.6%        |
> > | Mistral-7B Instruct v0.3 | 65.8%        | 39.5%        |
> >
> >
> > The Unconcerned:
> >
> > | Model                   | Violations@n | Completeness |
> > | ----------------------- | ------------ | ------------ |
> > | GPT-5                   | 34.9%        | 44.6%        |
> > | o3                      | 45.5%        | 43.1%        |
> > | GPT-4o                  | 20.3%        | 31.4%        |
> > | Gemini 2.5 Flash        | 56.9%        | 41.5%        |
> > | Llama-3.3 70B Instruct  | 52.4%        | 42.0%        |
> > | Qwen-3 32B              | 75.2%        | 47.2%        |
> > | Claude-4 Sonnet         | 51.6%        | 48.7%        |
> > | Mistal-7B Instruct v0.3 | 62.5%        | 36.3%        |

---

### Official Review · Reviewer_M7fA · 2025-11-01

**Soundness:** 3
**Presentation:** 3
**Contribution:** 4
**Rating:** 8
**Confidence:** 4

**Summary:**

The paper introduces CIMemories, a benchmark for evaluating whether memory‑augmented LLM assistants respect contextual integrity. The dataset uses synthetic user profiles (∼147 attributes per profile) and curated social contexts (∼46 per profile)

Each attribute–context pair is labeled (necessary to share vs. inappropriate to share) by sampling multiple “privacy personas” (Westin’s fundamentalist/pragmatist/unconcerned) from a strong LLM and retaining only unanimous labels; evaluation measures (i) Violation@n:worst‑case attribute leakage over n samples in contexts where the attribute should not be shared and (ii) Completeness—fraction of necessary attributes conveyed in contexts where they should be shared.

The paper further analyzes domain‑level failure modes (“granularity failure”), scaling effects, prompting defenses, and how both multi‑task composition and memory composition exacerbate leakage.

**Strengths:**

- Prior CI‑style evaluations (e.g., ConfAIde, PrivacyLens, CI‑Bench, LLM‑CI) largely focus on single‑shot vignettes or agent trajectories without rich, persistent user memory. CIMemories squarely targets that gap.
- The worst‑case attribute‑level Violation@n coupled with task‑level Completeness makes the privacy–utility trade‑off explicit; the visual breakdown in Figure 3 (p.6) convincingly illustrates “granularity failure” (right domain, wrong detail).
- Results span multiple model families and sizes and include simple defenses (prompting) and test‑time “reasoning” variants, yielding actionable observations (e.g., light reasoning sometimes lowers violations with minimal completeness loss)
- Well‑motivated by the trend from retrieval‑based memory (RAG/MemoryBank/MemGPT) toward long‑context, “prefix the memories” assistants, and shows those settings remain privacy‑fragile

**Weaknesses:**

- The “ground‑truth” labels (“necessary” vs. “inappropriate”) are produced by GPT‑5 with persona prompts, and then other LLMs are evaluated against those labels. Even with personas, this can encode the teacher model’s normative and stylistic biases. The paper argues LLMs are conservative vs. humans when labeling sensitive content, but no human audit is provided to calibrate false positives/negatives of the labeler on a subset. A 5–10% human‑labeled slice would materially increase credibility.
- Westin personas are U.S.‑centric and decades old; their priors may not reflect contemporary or cross‑cultural norms. The paper uses Westin‑based priors to mix persona votes but does not test sensitivity to those priors. A cross‑cultural variant or at least a sensitivity analysis is warranted.
- The headline “violations increase with more tasks/generations” is partly tautological because Violation@n is worst‑case per attribute over more trials. This is informative for risk, but the paper should also report the per‑turn hazard rate (probability of first leakage at turn t) and time‑to‑leak distributions to separate inherent risk from simple exposure.

**Questions:**

- Reference Missing In line 376. please fix
- Each context has ~7 “necessary” vs. ~84 “not‑to‑share” attributes, so a model that is slightly verbose can accumulate many apparent violations. Completeness is an average, whereas Violation@n is a worst‑case max across tasks and these aggregations are not symmetric. Consider reporting AU‑Privacy–Utility curves, pareto fronts or a balanced score.
- Reveal detection is too strict in one dimension and too lax in another. Could you use multiple judges in order to make it more robust. This under‑counts partial leaks (e.g., “my antidepressant dosage increased last month”) and leaks via implicature (e.g., “after the DUI class …”)
- Can you provide results under shared decoding parameters across all models? Right now, defaults differ by vendor and may be optimized for safety/verbosity differently.
- Given the heavy class imbalance (∼6.7 necessary vs. ∼83.7 not‑to‑share per context; Table 2), how would your conclusions change under a balanced per‑context metric (macro‑averaging) and a per‑turn hazard‑rate view rather than worst‑case Violation@n?

---

> ### Author Response · Authors · 2025-11-21
> **Response by Authors**
>
> We sincerely thank the reviewer for their comments and suggestions, and provide a detailed discussion below:
>
> >The “ground‑truth” labels (“necessary” vs. “inappropriate”) are produced by GPT‑5 with persona prompts, and then other LLMs are evaluated against those labels. Even with personas, this can encode the teacher model’s normative and stylistic biases. The paper argues LLMs are conservative vs. humans when labeling sensitive content, but no human audit is provided to calibrate false positives/negatives of the labeler on a subset. A 5–10% human‑labeled slice would materially increase credibility.
>
>
> As suggested, we perform a brief human validation study to compute agreement. We randomly sample a total of 100 attribute-context pairs (50 not-to-share, and 50 to-share as labelled by GPT-5), and ask 3 human raters (2 authors, 1 non-author) to label them. We then compute agreement between each rater and the model's labels, as well as between raters as below. Overall, we find that model predicts human labels respectably:
>
> |             | Model | Rater 1 | Rater 2 | Rater 3 |
> | ----------- | ----- | ------- | ------- | ------- |
> | **Model**   | -     | 85%     | 87%     | 83%     |
> | **Rater 1** | -     | -       | 83%     | 84%     |
> | **Rater 2** | -     | -       | -       | 80%     |
> | **Rater 3** | -     | -       | -       | -       |
>
> >Westin personas are U.S.‑centric and decades old; their priors may not reflect contemporary or cross‑cultural norms. The paper uses Westin‑based priors to mix persona votes but does not test sensitivity to those priors. A cross‑cultural variant or at least a sensitivity analysis is warranted.
>
> Indeed, a cross-cultural variant could include more representation, but Westin's work is some of the most reliable and repeated studies of personas, that is why we chose them. Eventually, as part of future work we also plan to add to the benchmark by collecting human labels from a diverse sample of the population as well, which can help improve cross-cultural representation.
>
> >Each context has ~7 “necessary” vs. ~84 “not‑to‑share” attributes, so a model that is slightly verbose can accumulate many apparent violations. Completeness is an average, whereas Violation@n is a worst‑case max across tasks and these aggregations are not symmetric. Consider reporting AU‑Privacy–Utility curves, pareto fronts or a balanced score.
>
> Our choice to focus on the worst-case for Violation@n is motivated by prior work, which emphasizes that even one failure can have severe consequences in privacy-critical settings [1,2]. For example, work in differential privacy often reports average case utility as a function of epsilon, which presents a measure of worst-case privacy. Regarding AU-Privacy-Utility curves, to draw such a curve requires tuning the variable that trades off the privacy and utility -  in our case, the main variable we have is the prompting strategy, which changes the verbosity of the model. We have run this for GPT-5 in Figure 5(c). We will certainly include it for more models in the final version. Depending on the minimum acceptable utility (completeness) threshold chosen for a downstream application, model rankings can change.
>
> [1] Martin, David J., et al. "Worst-case background knowledge for privacy-preserving data publishing." _2007 IEEE 23rd International Conference on Data Engineering_. IEEE, 2006.
>
> [2] Steinke, Thomas, and Jonathan Ullman. "The pitfalls of average-case differential privacy." _DifferentialPrivacy. org, July_ (2020).
>
> >The headline “violations increase with more tasks/generations” is partly tautological because Violation@n is worst‑case per attribute over more trials. This is informative for risk, but the paper should also report the per‑turn hazard rate (probability of first leakage at turn t) and time‑to‑leak distributions to separate inherent risk from simple exposure.
> >Given the heavy class imbalance (∼6.7 necessary vs. ∼83.7 not‑to‑share per context; Table 2), how would your conclusions change under a balanced per‑context metric (macro‑averaging) and a per‑turn hazard‑rate view rather than worst‑case Violation@n?
>
> Right now, a notion of per-turn hazard rate is best captured by figure 2, which shows how violations grow at a task-by-task granularity. We will also include these concrete numbers in the plot as annotated text. Regarding a balanced per-context metric such by averaging, one concern is that measuring privacy by an average case metric could potentially provide a sense of false security to users, since violations accumulate in reality. Per our discussion above, this is also the reason most prior work focuses on worst case as well. For example, considering GPT-5, computing via averaging over tasks would provide a metric score of 1.67%, which could appear low to users, even though many more than 1% of their attributes would eventually be violated as they engage with the model over time.

---

> ### Author Response · Authors · 2025-11-21
> **Response by Authors (cont.)**
>
> >Reference Missing In line 376. please fix
>
> We appreciate the pointer, and will fix the missing reference.
>
> >Reveal detection is too strict in one dimension and too lax in another. Could you use multiple judges in order to make it more robust. This under‑counts partial leaks (e.g., “my antidepressant dosage increased last month”) and leaks via implicature (e.g., “after the DUI class …”)
>
> We perform a brief human study to check the agreement of the judge with human raters. To this end, we randomly sample 50 attributes with responses where the judge marked them as revealed, as well as 50 attributes with responses where the judge marked them as not revealed. One author then independently judged these samples to determine if it was revealed, and we finally compute agreement with the judge. We find a **94%** agreement with the judge, suggesting that it performs similarly to a human. In general, our work does aim to provide a lower bound on leakage, and a more advanced adversary could perhaps draw more implicit inferences from the model response, which could lead to even more violations. This would present an excellent problem for future work.
>
> >Can you provide results under shared decoding parameters across all models? Right now, defaults differ by vendor and may be optimized for safety/verbosity differently.
>
>  We note that in some cases the vendor does not allow full control over sampling parameters, e.g., GPT-5 not allowing top-p, or even temperature values besides 1. This can make a systematic evaluation challenging. Below we evaluate open-source models on a common setting of temperature 1, while disabling top-p and top-k, to provide some initial measurements.
>
> | Model                    | Violation@5 | Completeness |
> | ------------------------ | ----------- | ------------ |
> | Llama-3.3 70B Instruct   | 45.5%       | 52.8%        |
> | Qwen-3 32B               | 74.0%       | 55.5%        |
> | Mistral-7B Instruct v0.3 | 56.8%       | 46.1%        |
>
> Drawing concrete takeaways from such experiments would likely need further investigation, since the space of decoding parameters is quite large. We would like to leave a full exploration to future work; it is likely that across all such strategies the Pareto frontier also changes.

---

### Meta-Review · Area_Chair_jXJq · 2026-01-10

**Summary:**

The paper introduces CIMemories, a benchmark that tests whether memory-augmented LLM assistants respect contextual integrity under flexible memory composition and multi-task composition. It formalizes two complementary metrics—Violation@n (worst-case leakage) and Completeness—and evaluates multiple model families, prompting strategies, and “reasoning” variants, revealing a clear privacy–utility trade-off and compositional failure modes.

Pros
1. Clear, timely problem framing focused on persistent, memory-augmented assistants rather than single-shot vignettes.
2. Well-specified benchmark construction and metrics that make the privacy–utility trade-off explicit.
3. Broad, careful analysis across models, sizes, and defenses; the “granularity failure” insight is useful for practitioners.

Cons
1. Ground-truth labels and the judge rely on LLMs; the human study is helpful but still small.
2. Personas and contexts are U.S.-centric; cross-cultural coverage and larger human audits remain future work.
3. Hazard-rate/time-to-leak reporting and stronger statistical testing are not fully developed.
4. Some mitigation guidance is preliminary; more prescriptive defenses would increase impact.

Overall, the benchmark is well motivated, clearly defined, and empirically illuminating. The rebuttal meaningfully strengthens validity and scope (added profiles, human agreement checks, persona-wise reporting), and the limitations are tractable for a camera-ready and follow-on work.

**Reviewer Concerns:**

1. Label quality and judge reliability: Added a small human validation (83–87% model–rater agreement) and a judge agreement check (~94% on a 100-item sample).
2. Benchmark scale: Expanded from 10 to 70 profiles; rankings remain consistent.
3. Unanimity filter transparency: Reported ~31% discarded pairs and domain-wise breakdown; added persona-wise results.
4. Comparison to prior work: Ran Confaide and showed 0% violations (42% completeness), indicating CIMemories surfaces distinct failure modes.
5. Decoding settings: Provided a shared-decoding subset for open-source models (noting vendor constraints for closed-source).
6. Mitigation insight: Analyzed “thinking” traces (e.g., explicit red-flag enumeration) and discussed why certain reasoning helps.
7. Metrics/visuals and nits: Will add AU-style exploration for verbosity/utility (shown for one model), annotate per-task growth in Fig. 2, fix missing reference and dangling “?”, and improve color choices for color-blind accessibility.

**Reviewer Scores:**

N/A

---

### Decision · Program_Chairs · 2026-01-26

Accept (Poster)